# Diagonal Tensile Test on Masonry Panels Strengthened with Textile-Reinforced Mortar

**DOI:** 10.3390/ma14227021

**Published:** 2021-11-19

**Authors:** Dragoș Ungureanu, Nicolae Țăranu, Dan Alexandru Ghiga, Dorina Nicolina Isopescu, Petru Mihai, Ruxandra Cozmanciuc

**Affiliations:** 1Faculty of Civil Engineering and Building Services, “Gheorghe Asachi” Technical University of Iaşi, 43 Mangeron Blvd., 700050 Iaşi, Romania; nicolae.taranu@academic.tuiasi.ro (N.Ț.); dan-alexandru.ghiga@student.tuiasi.ro (D.A.G.); dorina-nicolina.isopescu@academic.tuiasi.ro (D.N.I.); petru.mihai@academic.tuiasi.ro (P.M.); ruxandra.cozmanciuc@academic.tuiasi.ro (R.C.); 2The Academy of Romanian Scientists, 54 Splaiul Independentei, Sector 5, 050094 Bucuresti, Romania

**Keywords:** unreinforced masonry, textile-reinforced mortar, numerical and experimental study

## Abstract

This study presents the results of an experimental and numerical program carried out on unreinforced masonry panels strengthened by textile-reinforced mortar (TRM) plastering. For this purpose, five panels were constructed, instrumented and tested in diagonal shear mode. Two panels were tested as reference. The first reference panel was left unstrengthened, while the second one was strengthened by a traditional self-supporting cement mortar matrix reinforced with steel meshes. The remaining three panels were strengthened by TRM plastering applied on one or both faces and connected with transversal composite anchors. The numerical and the experimental results evidenced a good effectiveness of the TRM systems, especially when applied on both panel facings.

## 1. Introduction

Unreinforced masonry (URM) has a long history of being the predominant building technique in Romania and in Europe. A large share of this existing building stock consists of historical monuments, which have been designed to resist only gravitational loads or, in many cases, have not been designed at all, but simply erected according to a few generic rules of common practice [1,2,3,4,5,6,7]. In Romania, the regions where the masonry monumental buildings lie are all characterized by medium-to-high levels of seismic hazard. The composite nature of the bricks and of the mortar, the stocky arrangement of the URM walls and the almost zero tensile capability of the masonry material makes this structural system potentially vulnerable to seismic actions [8,9,10,11,12]. Consequently, most of the existing URM buildings require strengthening interventions works, to adapt the capabilities of the structural system to the requirements of the current seismic design codes. In order to design and perform a particular strengthening technique, the structural behavior and the possible failure modes of the unstrengthened structure should be well analyzed.

During seismic excitation, URM structures can develop two possible types of dominant failure modes, the in-plane shear or the out-of-plane bending mechanisms [13,14]. The second mechanism can be avoided by improving the overall stiffness of the structure with additional transversal connections [15]. On the other hand, the in-plane shear capacity governs the global seismic performances of the URM structures, since the lateral loads from the load bearing walls are transferred through the in-plane path to the foundations [16]. The in-plane dominant failure mechanism can be divided in several characteristic modes, including failure by sliding, rocking, toe crushing and shear. Due to the high weight of the URM buildings, the in-plane failure by sliding is rarely developed. The in-plane rocking and toe crushing failures are the most ductile ones. Nonetheless, the in-plane shear is a brittle and sudden-failure mechanism, being more severe and dangerous than the other in-plane modes. Moreover, the shear failure mode can be developed during moderate or even small earthquakes and its level of severity is influenced by a large variety of parameters, including bricks and mortar quality, load bearing walls thickness and arrangement, age of the structure, type of materials, existing seismic protection, soil conditions and modal characteristics of the URM structure. For these reasons, the development and improvement of shear strengthening techniques for URM structures have been topics of interest for many researchers.

Originally, the reinforced concrete strengthening techniques were adopted for URM elements as well [17]. These traditional strengthening techniques consist in stitching or filling the cracks and plastering the masonry element with a self-supporting cement mortar matrix reinforced with steel meshes. The main disadvantage of this approach consists in the high additional weight provided by the steel reinforcements. In order to overcome this issue, various fiber-reinforced polymer (FRP) composite elements have been developed as an alternative to steel reinforcements. Despite their advantageous features, such as high strength-to-weight ratio and ease of application [18,19,20], using FRP composite elements in strengthening applications for URM structures is usually associated with several possible limitations. These are the potential debonding of externally applied FRP strips, the incompatibility between most of the epoxy systems and many types of substrates (due to the large stiffness discrepancies) and the unsatisfactory behavior of common resin products at high and low temperatures. Recently, besides the conventional FRP composite strengthening systems, new materials have been developed and exploited for URM structures’ retrofitting. Among these materials, the textile-reinforced mortar (TRM) is considered to be highly beneficial since it has an improved compatibility with masonry substrates. This strengthening system consists in a layer of an inorganic matrix (usually a cement-based mortar) that is reinforced and applied on one or both faces of the URM walls. The reinforcements are textile fabric meshes made of fiber roving ply arranged on at least two orthogonal directions. The mechanical properties of the textile reinforcements are controlled by the amount of roving and their orientation, while the level of penetration of the mortar matrix is influenced by the mesh spacings [5,21].

According to previous studies [1,22,23,24,25,26,27], the TRM strengthening systems designed for URM structures are promising solutions that can provide significant increments in terms of shear capacity and pseudo-ductility both for in-plane and out-of-plane loading conditions. The in-plane structural behavior of the URM strengthened with TRM was studied for various wall types and configurations, including one-leaf or two-leaf brick arrangements [28,29,30,31] and tuff volcanic masonry walls [32,33]. The outcomes of some of the most relevant experimental programs related to the shear capabilities of URM walls strengthened with TRM are summarized in Table 1 [34]. The test types are abbreviated as follows: DG—diagonal test; CY—cycling test; SH—shear test. BR and VT stand for brick and volcanic tuff masonries, while the fiber meshes are denoted with S for steel, G for glass, C for carbon and B for basalt. The strength and the deformation ratios represent the capacities of the strengthened masonry specimens reported to the capacities of the URM ones.

By studying the data presented in Table 1, it can be observed that, for all the experimental campaigns, the improved behavior of the URM panels strengthened with TRM, in terms of strength and deformation capacity, has been validated. However, in some cases, the effective amount of reinforcement was not sufficient to provide significant increases in strength, but it enhanced efficiently the displacement capacity. The latter was proved to be valid even for TRM systems which consists of flax fabrics. In addition, when used as a component of TRM systems, these types of fibers do not evidence severe degradation of their mechanical properties even if they are subjected to various environmental exposures and aging protocols [35,36].

The experimental campaigns cited in Table 1 were aimed to characterize the structural behavior of the TRM systems either by testing samples made from fibers and mortar (tensile and adherence tests) or by addressing the global characterization of the systems by means of diagonal compression tests on reinforced masonry panels. By comparing these types of experimental results with the ones obtained by applying the available theoretical models, other studies reported several limitations of the theoretical models, related to the inability to consider combined or non-standard failure modes [43].

Moreover, despite the available studies carried out to investigate the shear structural behavior of the URM strengthened with TRM, which highlighted the effectiveness of this type of strengthening system, some important aspects have still not been fully investigated. The latter may refer to anchorage methods for masonry walls which enable intervention works on both faces (the side-to-side connection) or on a singular face (the middle-to-side connection); the increase in lateral capacity based on the type and configuration of the transversal connectors, pattern and effective length for various types of anchorages; the characteristic failure modes for the systems with additional reinforcement due to the radial ends of the anchorages; and the synchronization between the experimental and numerical analyses based on adapted modelling strategies.

This paper reports the results obtained through a numerical and experimental program that was carried out at the Laboratory of Heavy Structure, located at the Faculty of Civil Engineering and Building Services in Iași. The masonry materials and the strengthening systems, both traditional and modern, are rather common in Romania. In particular, five URM panels were constructed, instrumented and tested in diagonal shear mode. In addition, all the specimens were micro-modelled and subjected to a nonlinear finite element analysis using the Ansys Workbench software. Two panels were tested as reference. The first reference panel was left unstrengthened, while the second one was strengthened through a traditional self-supporting cement mortar matrix reinforced with steel meshes. The remaining three panels were strengthened by TRM plastering applied on one or both faces and connected with transversal composite anchors. After a detailed experimental set-up and testing procedures, the main experimental results and the key features of each configuration of the strengthening system are reported. In addition, the experimental results are compared to the ones obtained through a micro-detailed finite element analysis and a general good agreement is found.

## 2. Experimental Set-Up

### 2.1. URM Panel Specimens’ Configurations and Materials Properties

The experimental program was performed on five 1200 × 1200 × 115 mm URM panels. The specimens were manufactured using brick clay masonry units (Figure 1) and normal mortar of M15 class [44]. All of the URM panels were constructed by qualified masons to insure a proper level of craftmanship, similar to current practice. The general configuration of the specimens is presented in Figure 2.

The URM panels were left to cure for 28 days. Prior to strengthening and testing, the mechanical characteristics of the masonry material were determined and compared to the ones provided by the manufacturer. Bricks were the first component of the URM panels to be investigated.

Tests to determine the compressive strength were performed on five real-size specimens (Figure 3), which were loaded in compression in a Zwick Roell 1000kN universal testing machine, according to the specifications given in the Norm SR EN 772-1+A1 [45]. The value of the compressive strength provided by the manufacturer (15 MPa) was 28% lower compared to the average value that was determined experimentally (20.83 MPa). The computed standard deviation (σ = 0.45) indicates that the values tend to be close to the mean. Thus, it can be concluded that the experimentally determined values of the compressive strength of the brick units showed no statistically significant differences.

The aim of the micro, non-linear analysis that is presented in Section 3 of this work is to provide a numerical model for designing purposes. Thus, the compressive strength provided by the manufacturer (the conservative value) was adopted.

Flexural and compression tests were performed to evaluate mortar strength, according to EN 1015-11 [46] (Figure 4). The flexural tests were made on a mortar prism (mortar type S) with the nominal dimensions of 40 × 40 × 160 mm, which were subjected to three-point bending loading. Each of the two parts of the mortar prisms that resulted from the flexural tests were used for the compression tests. A total number of 15 mortar prisms were subjected to the flexural test and 30 specimens were tested in compression.

The results were determined under the assumption of general elastic–brittle behavior of the masonry wall loaded in tension. The average value of the tensile strength—f_m,t_—was approximately 3.66 MPa, while the average value of the compressive strength—f_m,c_—was 18.61 MPa. The computed standard deviations (σ_flexural_ = 0.80; σ_compresion_ = 1.18) indicate that the values tend to be close to the mean. Thus, it can be concluded that the experimentally determined values showed no statistically significant differences.

The experimental values were introduced as input data for the mechanical characteristics of the micro non-linear 3D model defined in Section 3 of this work.

As mentioned before, two masonry panels were manufactured and tested as reference (noted with URM and TSM). The first one (URM) was left unstrengthened, as shown in Figure 2. The second reference panel (TSM) was strengthened using a traditional system consisting of a self-supporting cement mortar matrix reinforced with steel meshes of ɸ6. The two faces of the strengthening systems were connected by means of steel connectors of ɸ6, passing through the URM panel and fixed at the exterior in the mortar layer. The overall configuration of the traditional strengthening system is illustrated in Figure 5. The mortar used for this strengthening approach has identical characteristics with the one that was used for the URM panel assemblage.

The remaining three URM panels were strengthened by plastering on one face (for one specimen—TRM1) or on both faces (for two specimens—TRM2 and TRM3) with a two-component ready-mixed, high ductility pozzolan-reaction mortar reinforced with an alkali-resistant (AR), pre-primed glass fiber mesh (Figure 6) [47,48].

According to the manufacturer [47], the mortar was cement-based, made by blending special admixtures and synthetic polymers in water dispersion and reinforced with glass fibers. The physical and the mechanical characteristics of the strengthening mortar as provided by the manufacturer are listed in Table 2. This mortar has adequate adhesion strength and, when hardened, it forms a compact and uniform layer that is impermeable both to water and aggressive atmospheric gases, while remaining highly permeable to water vapors. In total, 15 mortar prisms were prepared and subjected to the flexural test and the resulted 30 specimens were tested in compression (Figure 7). The average experimental value of the tensile strength—f_m,t_—was approximately 4.35 MPa, while the value given in the technical data sheet was equal with 8 MPa. Similarly, the average experimental value of the compressive strength—f_m,c_—was 19.77 MPa, while the producer indicated a minimum 25 MPa strength in compression at 28 days. The computed standard deviations (σ_flexural_ = 0.14; σ_compresion_ = 1.21) indicate that the values tend to be close to the mean. Thus, it can be concluded that the experimentally determined values showed no statistically significant differences.

According to the technical data sheets, the producer determined the compressive and the flexural strength according to the provisions given in the Norms EN 12190 and EN 196/1 [49,50], while the experimental values mentioned above were determined according to the specifications given in the Norm EN 1015-11 [46]. For the micro non-linear 3D model, the mechanical properties were defined according to the experimentally determined values.

The textile meshes used to reinforce the mortar are part of the same custom design system for masonry strengthening. The meshes consist in multiple dry carbon high-strength fiber roving disposed in a square pattern [48]. Due to the custom configuration of the weave, the mechanical properties of these meshes are controlled so that, when applied on URM walls together with the mortar, they make up for the masonry’s lack of tensile strength and increase the overall ductility so that the shear stresses are distributed more evenly. As a result, in the event of seismic actions, the TRM strengthening system has the ability to evenly distribute the stresses and the strains over the entire surface of the URM walls, so that the dominant failure mechanism is shifted from a fragile type to a ductile one. Moreover, it should be noted that this type of TRM system has potential advantages when used on masonry historical buildings. In this frame, the system works in parallel with the existing structural elements, without influencing the distribution of the masses and the rigidity within the structures. The latter is a significant gain, particularly for seismic design, where stresses are proportional to the masses involved. The physical and mechanical properties of the mesh are listed in Table 3 [48].

As it can be observed in Figure 6, the face/faces of the TRM strengthening systems were additionally fixed with four transversal composite connectors, located on the corners, so as to ensure an improved connection to the masonry substrate and to avoid the out-of-plane failure. The composite connectors were unidirectional carbon fiber threaded inside a gauge wrapping [51]. Due to this appearance, this type of composite connectors is also referred to as cords. The physical and the mechanical characteristics of the cords, as given by the manufacturer, are listed in Table 4. The configuration of the anchors was selected according to the specification given by the manufacturer [51].

In order to achieve a full interaction among the cords and the specimens, a chemical anchor was manufactured by means of an adhesive product. A bi-component styrene-free adhesive made from polyester resin was poured so as to fill the holes through which the cords were inserted. The physical and mechanical characteristics of the adhesive, as provided by the manufacturer, are listed in Table 5 [52].

### 2.2. Construction Process

The URM panels were constructed of bricks with nominal dimensions of 1200 by 1200 by 115 mm. The latter were assembled by an experienced mason using a mortar made of Portland cement and sand, in the proportion and quality that are commonly used in the traditional Romanian masonry buildings (one part Portland cement to three parts sand). The thickness of both the horizontal and vertical mortar joints was 10 mm. Since the bricks were stored in indoor conditions, each unit was wetted before usage. After all the specimen’s manufacturing processes were complete, the surfaces were cleaned of dust, pollutants or loose materials by sandblasting and low-pressure water jetting.

For the second reference panel, two steel meshes of ɸ6 were disposed on each side (Figure 8). The meshes were tied together by means of transversal steel connectors of ɸ6. The attachments between the meshes and the transversal connectors were achieved by means of thin binding steel wires. Next, the surfaces of the panel were wetted and the mortar layer was overlaid so as to cover all the reinforcements. The overall thickness of the reinforcing layer was around 12 cm.

The remaining URM panels were strengthened with TRM systems. First, all the surfaces were wetted and the first layers of the strengthening mortar were overlaid. The two constituents of the mortar, component A (powder) and component B (liquid), were mixed according to the specifications of the manufacturer [47]. The final material consisted of a plastic–thixotropic blend that was applied to the panels’ surfaces with a flat metal smoothing towel.

After the first layer of the mortar was applied and while it was still wet, the meshes were placed over the surfaces of the panels. The meshes were lightly pressed down with a towel for them to adhere to the mortar. The adjacent pieces of the meshes were overlapped both widthways and lengthways by approximately 15 cm. Next, the second layer of the mortar was applied. The width of both mortar layers was approximately 5–6 mm.

In order to reduce the occurrence and development of the de-bonding phenomena and to improve the static efficiency of the system, the faces of the TRM system were tied with composite connectors [51]. For this purpose, after the mortar was cured, holes were drilled in the substrate and the loose material was removed with compressed air. Additionally, the surfaces inside the holes were also cleaned with a long-bristled brush. The protective gauze from the cords was unrolled to a length equal to the depth of the holes that were previously drilled into the panels. Before the cords were inserted into the panels, the adhesive was extruded, starting from the bottom of the holes, until the spaces were fully filled. The two components of the adhesive were automatically mixed together when the product was extruded, using a static mixer that was supplied together with the cartridges. The cords were then inserted into the adhesively filled holes and the excessive material was removed. The last remaining protective gauzes were removed and the fibers were spread over the substrate in a circular pattern. The overall diameter of the circle was approximately 25 cm. Finally, the spread fibers were embedded in a layer of mortar.

It is worth mentioning that, in the case of the URM panel strengthened only on one face, the anchoring was performed starting from the middle of the element to its face. This simulated a common situation for monumental buildings, when strengthening system needs to be applied without imposing any interventions to one side of the wall. The construction stages of the TRM1, TRM2 and TRM3 specimens are illustrated in Figure 9.

### 2.3. Instrumentation and Testing Procedures

The tests were performed according to the specifications given in ASTM E519/E519M—15 [53]. The latter describe a test procedure designed to determine the shear strength of masonry panels with modular dimensions of 1.2 by 1.2 m, by loading them in compression along the diagonal. Due to the loading direction, a diagonal tension failure is induced within the specimen. Thus, the specimens were carefully lifted, rotated at 45^0^ and positioned on the transportation carriage in such a way that no disturbance was caused either to the unstrengthened or the strengthened panels (Figure 10).

The load was applied to the URM panel through a steel shoe placed at the top corner and transmitted to a similar shoe at the bottom corner (Figure 11). Each loading shoe had a loading area of 330 mm by 250 mm. A hydraulic jack with an overall capacity of 500 kN was incorporated between the clamp of the testing machine and the top loading shoe. The deformation of the specimens, elongation of diagonals and compression were monitored by four linear variable differential transducers (LVDTs), with two being placed on each face of the panel, oriented perpendicular and parallel to the loading direction (Figure 11). All the data were captured and stored using a data acquisition system (DAQ). Each test was load-controlled; thus, the load was increased monotonically at a constant speed of approximately 0.5–0.6 kN/s.

## 3. Numerical Approach

### 3.1. Introduction

Masonry is an anisotropic composite material which consists of masonry units (bricks, stone, etc.) assembled with or without mortar. The mechanical models that were developed to evaluate the structural behavior of the URM structures fall under the category of no-tension material models. From the theoretical point of view, URM walls are considered to behave as a linear elastic material when subjected to compressive stresses. However, based on the mechanical and elastic properties of the components, the mechanical model and, by default, the numerical one may be adapted from a typical linear–elastic to an elastic–plastic one.

Generally, three distinct modelling approaches can be adopted to simulate the structural behavior of a URM wall loaded in shear [54,55,56,57,58,59,60,61,62,63,64,65]. The first one, referred to as macro-modelling, consists in a continuum homogenous element that does not distinguish between the masonry units and mortar or between the individual components of the strengthening system. The macro-model is not suitable for the analysis of the URM walls strengthened by TRM plastering since, in this case, the stresses tend to converge into a narrow region along the faces of the composite transversal connectors. More specifically, the numerical macro-models are unable to account for the micro-mechanical characteristics at the interface levels. The second approach, referred to as the simplified micro-modelling, consists in a combination between micro and macro modelling techniques. The numerical model obtained by applying this approach has both continuum elements (for masonry units and mortar) and discontinuous elements (for the interface levels). This simplification can significantly reduce the computational costs. However, when the strengthening system is made of multiple materials with various mechanical and elastic properties (e.g., reinforced mortar, textile cords, reinforcing meshes, adhesives, etc.), the simplified numerical model can undervalue the overall shear response of the strengthened URM wall. On the other hand, the detailed micro-numerical approach allows us to identify and monitor the complete stress–strain state of the strengthened URM walls. These models are designed to account for each component of the physical models and for the particular properties of each material, thus reflecting the shear structural behavior of the URM walls strengthened with TRM in a more realistic manner.

### 3.2. Model Definition

In this study, a detailed micro non-linear 3D model was developed to accurately simulate the laboratory test conditions (Figure 12). The non-linear numerical analyses were carried out using the commercial, multi-purpose finite element software Ansys V 15.0, Ansys, PA, USA [55,56,57,58,63,64].

For the numerical 3D micro-models, the bricks and the mortar were simulated using SOLID187 tetrahedron finite elements (Figure 13a) [55]. The latter consist of higher-order 3D elements with 10 nodes, each of them defined by three degrees of freedom. These elements were selected based on their characteristic quadratic displacement behavior which can account for large deflections and significant strain variations. The components of the TRM system and the composite cords were modelled by SHELL 63 finite elements, as they possess both membrane and bending capabilities (Figure 13b). Moreover, the 3D micro-models were defined to account for three distinct interface levels (bricks–mortar, TRM–bricks, TRM–mortar). CONTA 174 bonded contact elements were modelled at each interface level so as to detect any possible contact separation between the TRM system and the URM panels (Figure 13c).

The bricks were assumed to behave as linear elastic isotropic materials and the plasticity properties were entirely assessed to the mortar, which was modelled as an elastic–plastic material. In this approach, the structural characteristics of the mortar are capable to account for the global non-linear behavior of the URM module. In addition, by applying this modelling strategy, it is assured that failure occurs in a characteristic mode, which is defined by a continuous diagonal crack that passes through the head and the bed mortar joints. The plasticity of the mortar was defined using the Drucker–Prager formulation. According to the latter, the material is characterized by a specific domain of yield stresses. When the yield surface is reached, the mortar starts to deform plastically and, upon further increase in load, the plastic flow is initiated.

As mentioned before, the complete test specimens were modelled, including the interface levels. For this purpose, the augmented Lagrange formulation was used to define the contacts. The penetration points were determined for both bricks and mortar based on the integration point detection.

Since the stresses vary with a high gradient between distinct components of the 3D model, the mesh size was set differently for bricks, mortar, TRM and composite cords. For the bricks, the maximum mesh size was limited to 30 mm; for the mortar and the composite cords, the mesh size was limited to maximum 2 mm; whereas, for the TRM system, a maximum length of 80 mm was imposed in the model (Figure 14).

The nonlinear 3D micro-detailed models were analyzed by using the Newton–Raphson iteration method. More specifically, the method was applied by gradually increasing the load, up to the limit where the convergence criterion was satisfied. Furthermore, whenever required, various sub-steps were defined so as to divide the load ratio. Thus, the final step/sub-step corresponded to the ultimate load. This value was reached at the end of the elastic branch, immediately before an abrupt change in displacement was recorded. This change also indicated that the convergence criterion was no longer satisfied.

## 4. Results and Discussion

In the case of the URM reference panel, the failure occurred in the mortar layer by splitting the specimen along the horizontal and vertical joints (Figure 15). Besides the typical propagation path of the crack (through the mortar layer), shortly before the force reached its ultimate value, the crack propagated even through the bricks.

By analyzing the load–displacement curve (on both faces of the panel) of the URM reference panel, depicted based on both experimentally and numerical results, it can be observed that, in the first stage, the specimen’s response was quasi-linear–elastic with almost constant stiffness value. In the second plateau of the graph, the load–displacement curve illustrates a plastic phase which represents the degraded stiffness state of the URM panel (Figure 16). The plastic behavior was recorded immediately after the first cracks had started to develop. At the end of the plastic branch, all the cracks indicated by arrows in Figure 15 were joined in a continuous crack that spanned, in the vertical direction, between the loaded corners of the masonry panel. At the end of the test, the masonry panel was separated into two parts along the direction of the crack.

As it can be observed, the numerical and the experimental results are in good agreement both for ultimate values and distributions. It should also be mentioned that a sign convention was adopted in order to represent, in the same graph, both the horizontal and vertical displacements. For the evaluation of the shear stress–shear strain state, all the data were taken as absolute values.

In the case of the TSM panel, even though the overall thickness of the strengthening systems is around 12 cm, the typical pattern for crack development (across the masonry mortar joints) can be easily identified (Figure 17). When the load reached its ultimate value, large surfaces of mortar started to expel, exposing the steel reinforcement meshes (Figure 17b). Moreover, the failure by panel splitting is evidenced by the distribution of the horizontal displacement, where it can be observed that, at a relatively constant load, or even slightly decreasing, the displacements increase largely. Two local failures were also recorded in the ranges 38–42 KN and 50–52 KN, respectively (Figure 18).

Unlike the reference URM panels, the TRM-strengthened ones (TRM1, TRM2 and TRM3) evidenced a gradual failure mode, which is highly advantageous for the masonry structures located in earthquake-vulnerable areas. For the TRM1 panel, the crack pattern is similar to the ones developed for the reference panels (Figure 19a). However, on the plastered face, the crack pattern is slightly shifted toward the borders of the circular regions where the composite connectors were embedded with the mortar (Figure 19b).

The TRM 2 and TRM3 panels developed a particular crack pattern which followed the border line of the composite anchors (Figure 20). Besides preventing the de-bonding phenomena and improving the static efficiency of the TRM system (the main functions of the composite anchors), the exterior part of the cords provided an additional reinforcement for the system. More specifically, the radial orientation of these fibers modified the crack pattern by shifting its original vertical direction towards the border region of the exterior side of the composite anchors (Figure 20b,c). Moreover, the sliding effect at the joints was negligible due to the presence of the end parts of the anchors (also referred as *diatones*). On the contrary, in the case of similar experimental works, that were performed on masonry panels tied with punctual transversal connectors, the sliding phenomena was more intense, as reported in [66].

As it can be observed in Figure 21, Figure 22, Figure 23, Figure 24 and Figure 25, beside the linear–elastic plateau, for all the specimens (including the URM panel strengthened only on one face), a characteristic plateau of constant load and increasing displacement was recorded. This plateau is similar to the plastic range (yield plateau) representation of structural carbon steel elements. As shown in the load–displacement curves, in the plastic stage, the load remains constant while the displacements continue to increase, meaning that the elongation and contraction continue as long as the load is not removed. In addition, this behavior indicates that the TRM systems provide significant additional ductility that enables the URM panels to largely deform before failure. Nevertheless, this behavior demonstrates the efficiency of the TRM systems that are applied only on one face of the panel, a case which often comes across in the strengthening applications of monumental masonry buildings. The anchoring techniques (both the side-to-side and the middle-to-side method) provide sufficient lateral capacity to avoid the possible out-of-plane failure.

The shear stress–shear strain distributions of all panels are presented in Figure 24 and Figure 25. According to ASTM E519/E519M—15 the shear stress is computed using Equation (1) [53].
(1)$$S_{s}=\frac{0.707P}{A_{n}}$$
where S_s_—shear stress (MPa); P—load measured along the diagonal pattern; A_n_—net area of the panel; $A_{n}=\frac{w+h}{2}t\times n;$ w—width of the panel (mm); h—height of the panel (mm); t—thickness of the panel; n—the percentage of the gross area that is solid (expressed as a decimal).

According to ASTM E519/E519M—15, the shear strain is computed using Equation (2) [53].
(2)$$Ɣ=\frac{\Delta V+\Delta H}{g}$$
where Ɣ—shear strain (mm/mm); ΔV—shortening on vertical direction; ΔH—extension on horizontal direction; g—monitoring length.

As it can be observed in Figure 24 and Figure 25, for all the specimens, the shear stress–shear strain distribution curves start with a relatively steep slope and increase linearly until the beginning of the plastic range.

In the cases of the URM panel and of the traditional, strengthened panel (URM and TSM panels), in the plastic phase, a substantial degraded stiffness can be observed. On the other hand, in the cases of the TRM-strengthened panels (TRM1, TRM2 and TRM3), the deformation capacity is considerably improved and the characteristic (yield plateau) that was previously mentioned for the load–displacement distribution can be easily identified. Thus, TRM systems provide significant additional ductility that enables the shear stresses to remain constant while the shear strains continue to increase, meaning that the elongation and contraction continues as long as the load is not removed.

The maximum shear stress recorded for the URM panels strengthened on both faces (TRM2 and TRM3) were 20% higher than the one measured for the URM panel strengthened only on one face. When it comes to the numerical values, similarly, an increase of 33% was recorded for the maximum shear stress for the panels strengthened on both faces when compared to the one of the panel strengthened on one face. The experimentally obtained value for the shear strain was 23% higher for the TRM2 and TRM3 specimens when compared to the one measured for the TRM1 specimen. However, the numerical values of the shear strains were almost equal in both cases.

The ultimate values of both the shear stress and shear strain that were experimentally recorded for the TSM panel were almost equal to the ones recorded for the TRM2 and TRM3 panels. However, the structural improvement of the TSM panel was obtained by increasing the specimen overall thickness by approximately 12 cm, while the overall thickness of the TRM2 and TRM3 panels was increased by only 2 cm. Thus, the total weight of the traditional strengthened system was considerably higher than the one of the modern TRM system.

According to ASTM E519/E519M—15 [53], the modulus of rigidity—G—is computed as the ratio between the shear stress—S_s_—and the shear strain—Ɣ—[53]. The stiffness of the URM panels can be evaluated by quantifying the modulus of elasticity—E. The latter can be calculated using Equation (3), where υ was adopted as 0.25, as reported in previous studies relating to similar URM panels [14,60].
(3)$$E=2G\left(1+\upsilon \right)$$

The ductility of the URM panels can be quantified by the drift ratio—δ—Equation (4).
(4)$$\delta_{u}=\frac{\Delta u}{H}$$
where Δu—the displacement measured along the diagonal, corresponding to the ultimate load; H—height of the URM specimens.

The experimental and the numerical results are listed in Table 6.

As presented in Figure 21, Figure 22, Figure 23, Figure 24 and Figure 25, the response of all the specimens was non-linear and the post-peak behavior in the cases of the strengthened panels was characterized by a relatively gradual strain softening. The shear stiffness of the TSM panel, represented by the modulus of rigidity (G), increased by 34%, when compared to the experimentally recorded value for the URM panel. Similarly, the numerical results show an increase by 43% from the previously mentioned values. In the case of the TRM1 panel, the shear stiffness was increased by 36% (experimentally determined value) and 31% (numerically determined value), when compared to the values determined for the URM panel. The largest increase in the shear stiffness was recorded for the TRM2 and TRM3 panels (approximately 42% with respect to the experimental value and 62% with respect to the numerical value).

The ductility of the URM panel, expressed as the drift ratio, is considerably lower when compared to the values determined for the strengthened panels. Thus, increases in ductility of 64% for the TSM panel, 54% for the TRM1 panel, 66% for the TRM2 panel and 34% for the TRM3 panel were recorded with respect to the experimentally determined value of the URM panel. Similarly, increases in ductility of 73% for the TSM panel and 67% for the TRM1, TRM2 and TRM 3 panels were recorded with respect to the numerically determined value of the URM panel.

## 5. Conclusions

This paper presents the main outcomes of a numerical and experimental study which relates to the structural efficiency of a modern strengthening technique designed for URM panels. The latter consists of a TRM jacketing that can be applied either on one side of the wall or on both sides. The structural TRM layers can be anchored both by the side-to-side method and the middle-to-side one.

From the findings summarized above, the following can be concluded:The observed damaged patterns both for the URM and TSM specimens matched the characteristic crack path, which consisted in a continuous-propagation step-like line that spread along the compressed diagonal, through the horizontal and vertical mortar joints.The TRM2 and TRM3 panels developed a particular crack pattern which followed the border line of the composite anchors.The TRM systems evidenced a general and significant increase in the original shear capability and ductility of the URM panel. They also enabled the URM panels to largely deform before failure.The selected type of anchorage system provided sufficient lateral capacity to avoid the possible out-of-plane failure.The ultimate loads that were experimentally determined increased by 53% for the TSM panel, 41% for the TRM1 panel, 56% and 57% for the TRM2 and TRM3 panels, when compared to the ultimate load recorded for the URM panel.Similarly, the numerical results evidenced an increase of 56% for the ultimate load of the traditional strengthened panel, 45% for the panel plastered with TRM on one face and 57% for the 3D micro-model with TRM jacketing on both faces, when compared to the ultimate load recorded for the URM reference panel.The ultimate values of the mechanical properties that were experimentally and numerically investigated were almost equal for the TSM panel and the TRM2 and TRM3 panels.The numerical and the experimental results validate the efficiency of the TRM systems that are applied only on one face of the wall, a case which often comes across in the strengthening applications of monumental masonry buildings.

Although further experimental studies are required for an adequate assessment of the investigated strengthening techniques (e.g., position, dimension and number of the composite anchors) and for the validation of the micro-numerical model, the studies discussed in this paper represent a background for the general structural efficiency of the TRM strengthening systems.

## Figures and Tables

**Figure 1 materials-14-07021-f001:**
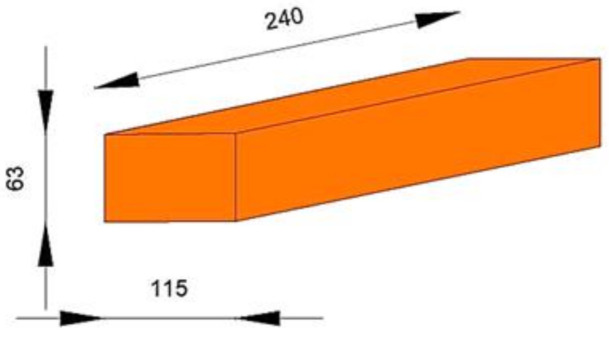
Brick unit (dimensions in mm) [44].

**Figure 2 materials-14-07021-f002:**
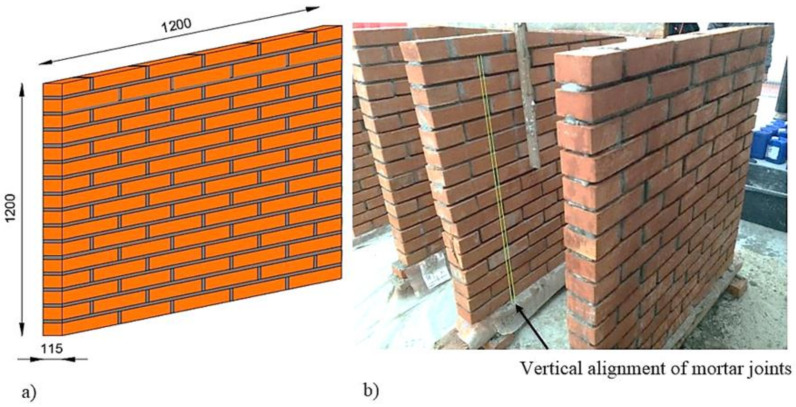
Specimens: (**a**) geometric configuration (values in mm) and (**b**) URM craftmanship.

**Figure 3 materials-14-07021-f003:**
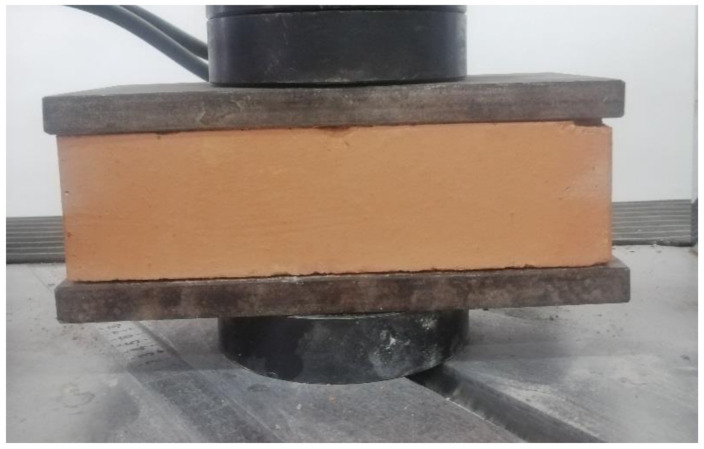
Brick unit tested in compression using a WAW 600 Universal Testing Machine.

**Figure 4 materials-14-07021-f004:**
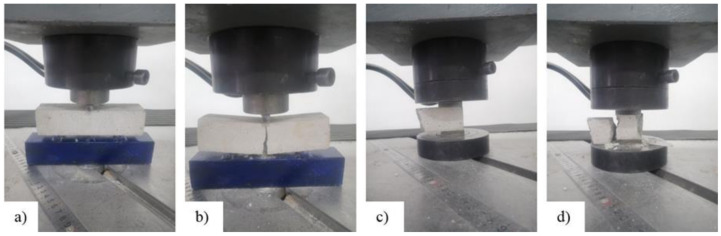
Flexural test: (**a**) mortar prism specimen and (**b**) failure mode. Compression test: (**c**) specimen and (**d**) failure mode.

**Figure 5 materials-14-07021-f005:**
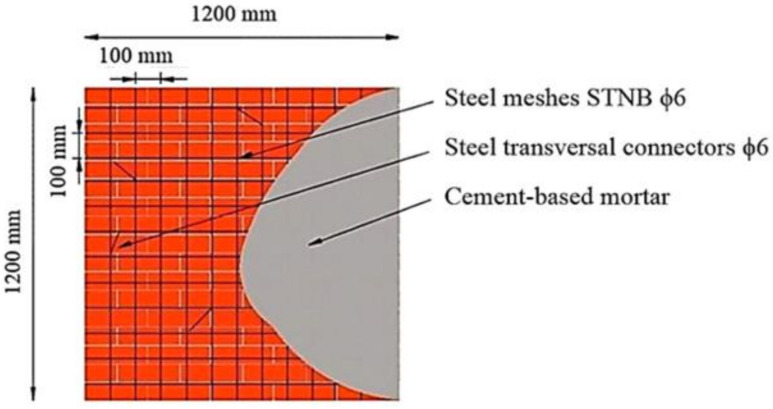
Configuration of the traditional strengthening system—TSM panel.

**Figure 6 materials-14-07021-f006:**
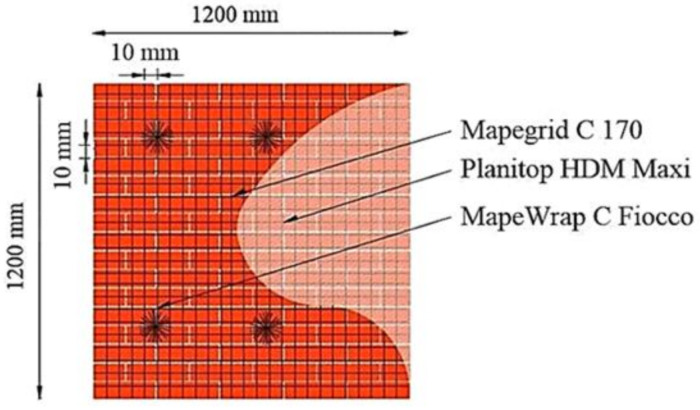
Configuration of the TRM strengthening system.

**Figure 7 materials-14-07021-f007:**
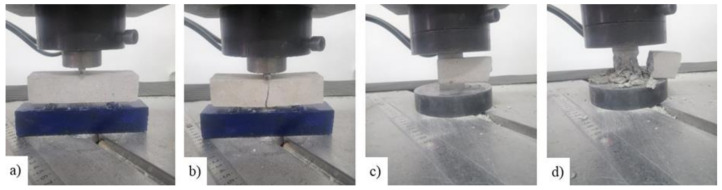
Flexural test: (**a**) mortar prism specimen and (**b**) failure mode. Compression test: (**c**) specimen and (**d**) failure mode.

**Figure 8 materials-14-07021-f008:**
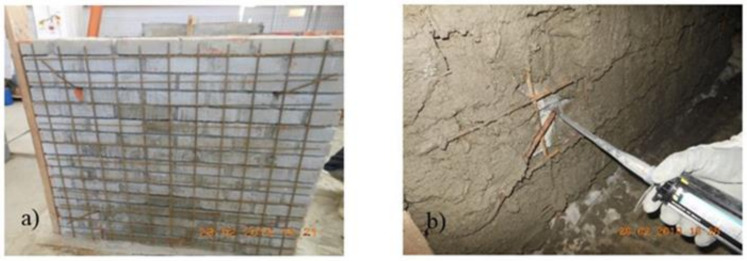
(**a**) TSM panel and (**b**) transversal connector.

**Figure 9 materials-14-07021-f009:**
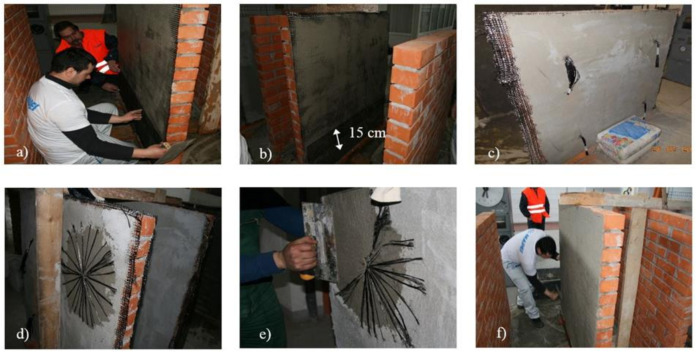
Construction stages: (**a**) application of the first layer of mortar and reinforcing meshes; (**b**) overlap of the reinforcing meshes; (**c**) insertion of composite cords; (**d**) cords spreading in a circular pattern; (**e**) embedment of the cords with mortar; (**f**) surface levelling and final checks.

**Figure 10 materials-14-07021-f010:**
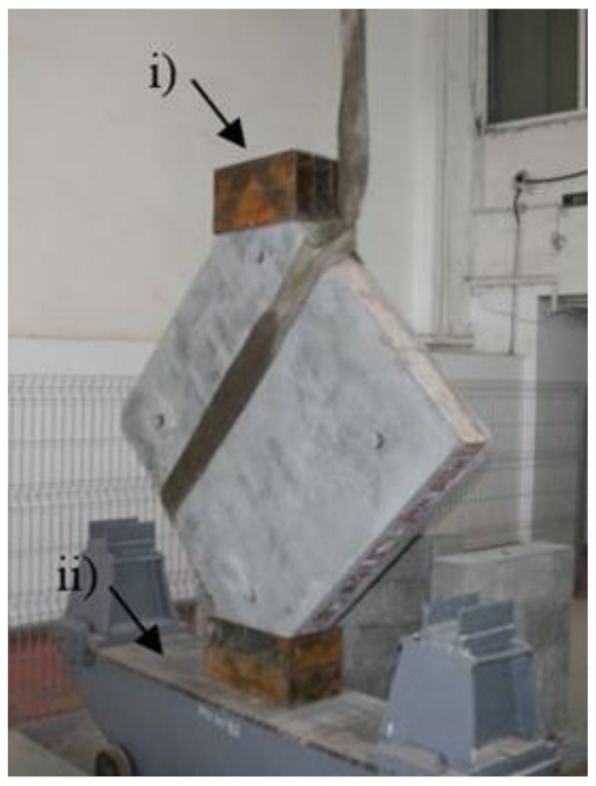
Specimen positioned on the transportation carriage: (i) loading shoe and (ii) transportation carriage.

**Figure 11 materials-14-07021-f011:**
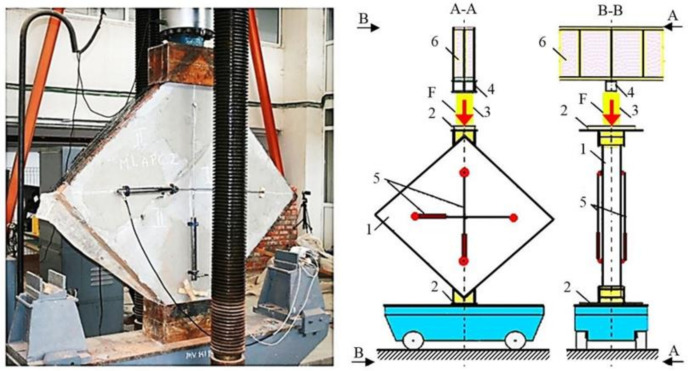
Specimen instrumentation and loading conditions: 1—URM wall; 2—loading shoe; 3—loading jack; 4—loading cell; 5—LVDTs; 6—steel channel.

**Figure 12 materials-14-07021-f012:**
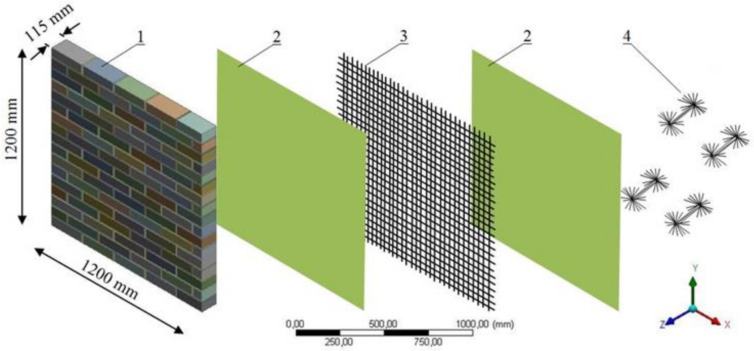
Three-dimensional model: 1—URM panel; 2—Planitop HDM Maxi mortar; 3—Mapegrid C 170 meshes; 4—MapeWrap C Fiocco composite connectors.

**Figure 13 materials-14-07021-f013:**
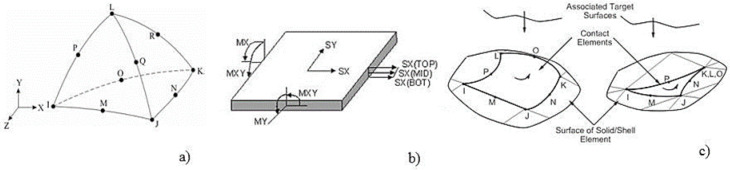
Finite elements: (**a**) SOLID187, (**b**) SHELL 63 and (**c**) CONTA174. *i, j, k, l, m, n, o, p, q, r—nodes; MX, Y, XY—bending moments; SX, Y—displacements. Figure adapted from [59].

**Figure 14 materials-14-07021-f014:**
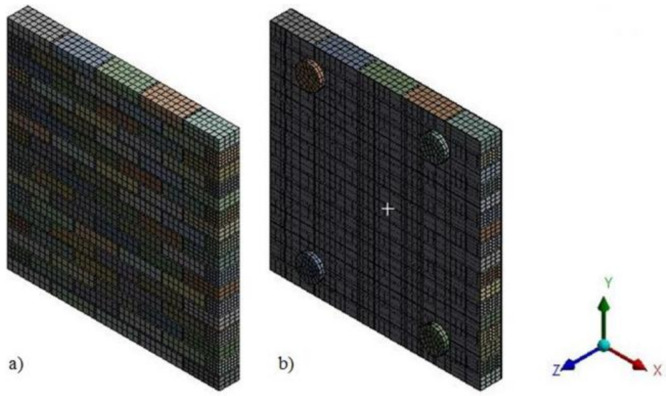
Finite element mesh of (**a**) URM panel and (**b**) URM panel strengthened with TRM.

**Figure 15 materials-14-07021-f015:**
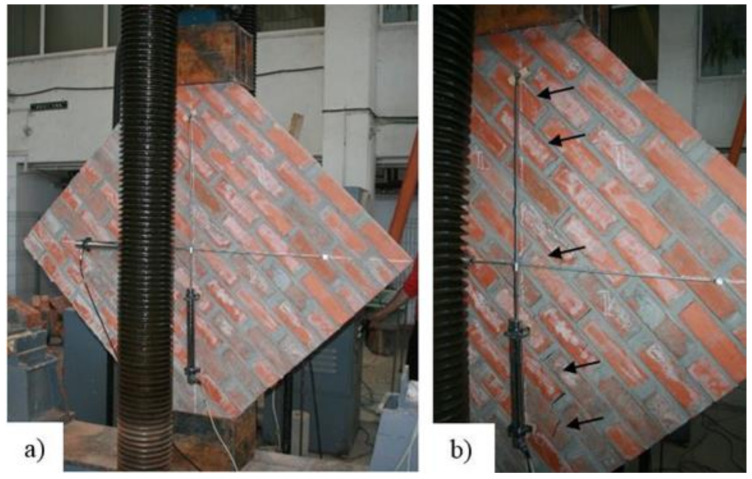
(**a**) URM panel loaded into the testing machine; (**b**) Failure mode—crack propagation pattern through the mortar joints.

**Figure 16 materials-14-07021-f016:**
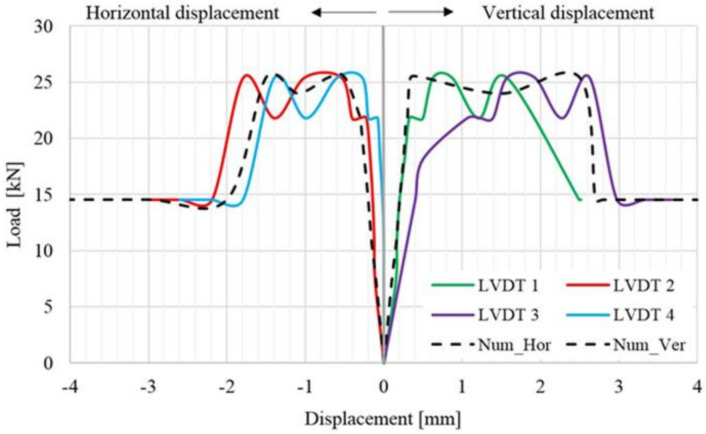
Load–displacement distribution for the URM panel.

**Figure 17 materials-14-07021-f017:**
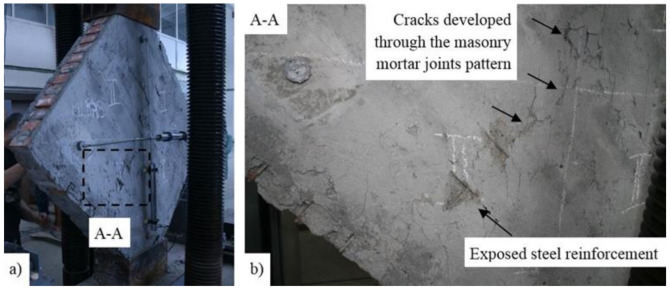
(**a**) Failure mode of the TSM panel and (**b**) detail of A-A.

**Figure 18 materials-14-07021-f018:**
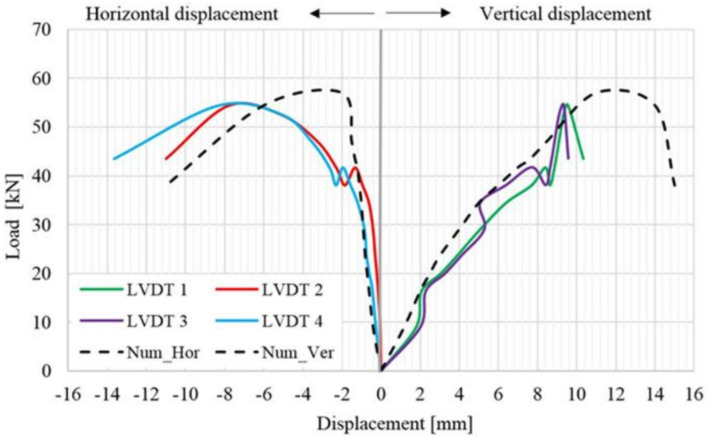
Load–displacement distribution for the TSM panel.

**Figure 19 materials-14-07021-f019:**
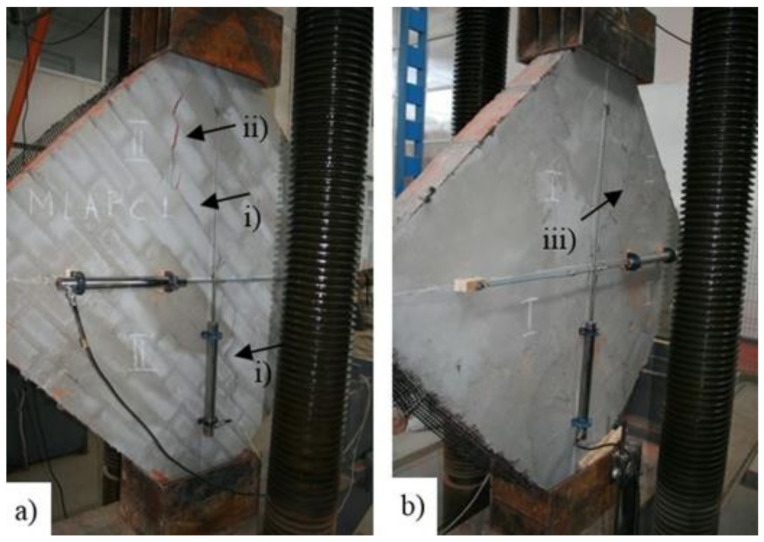
URM panel strengthened with TRM on one face (TRM1 panel). (**a**) Unstrengthened face: (i) crack propagation through mortar joints and (ii) crack propagation through brick units; (**b**) Strengthened face: (iii) crack propagation close to the composite anchors.

**Figure 20 materials-14-07021-f020:**
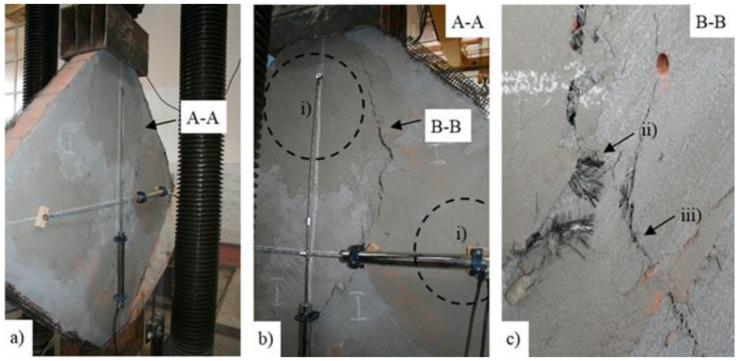
TRM2 and TRM 3 panels. (**a**) Crack pattern. (**b**) Detail of A-A: (i) exterior side of the composite anchors. (**c**) Detail of B-B: (ii) exposed composite cords and (iii) failure of the TRM by fiber mesh rupture.

**Figure 21 materials-14-07021-f021:**
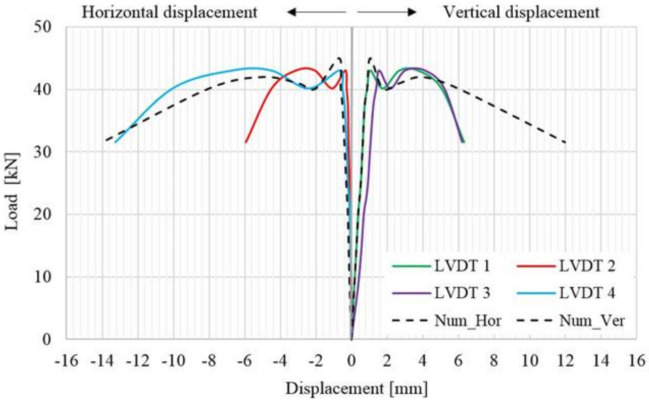
Load–displacement distribution for the URM panel strengthened with TRM on one face (TRM1 panel).

**Figure 22 materials-14-07021-f022:**
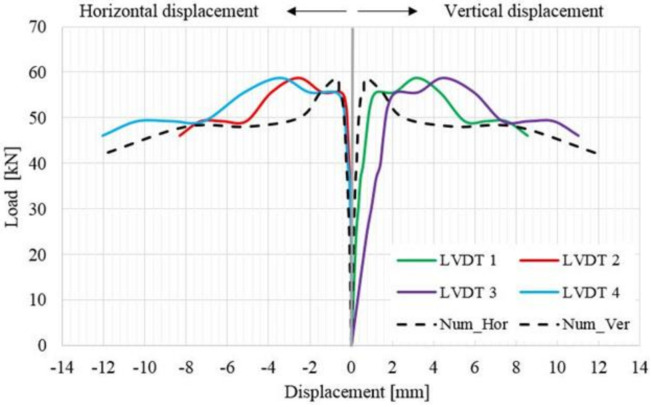
Load–displacement distribution for the URM specimen strengthened with TRM on both faces (TRM2 panel).

**Figure 23 materials-14-07021-f023:**
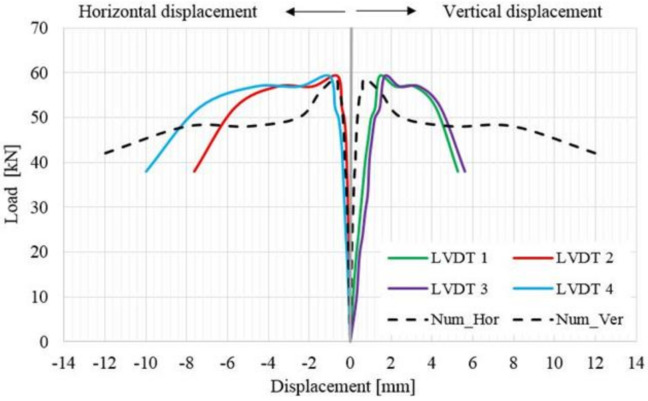
Load–displacement distribution for the URM specimen strengthened with TRM on both faces (TRM3 panel).

**Figure 24 materials-14-07021-f024:**
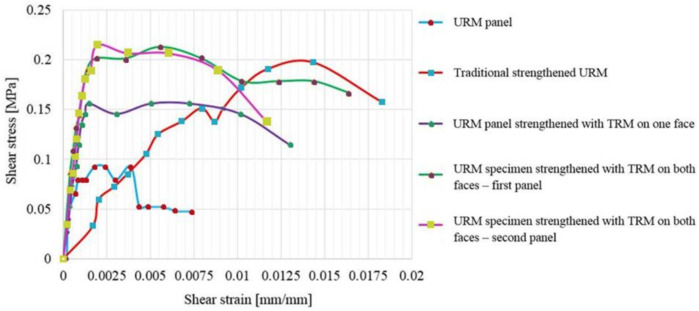
Shear stress–shear strain distributions: experimental results.

**Figure 25 materials-14-07021-f025:**
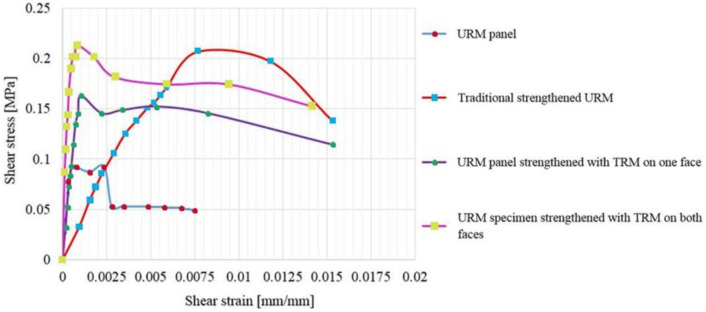
Shear stress–shear strain distributions: numerical results.

**Table 1 materials-14-07021-t001:** In-plane tests of URM panels strengthened with TRM [34].

Authors	Test Type	No. of Panels	Type of Masonry	Type of Reinforcing Meshes	No. of Strengthened Sides	Failure Mechanism	Strength Ratio	Deformation Ratio
Prota et al., 2006 [33]	DG	4	TF	G	1	Tensile rupture	1.7	2.4
Prota et al., 2006 [33]	DG	4	TF	G	2	Crushing	2.2	2.3
Papanicolaou et al., 2007 [37]	SH	4	BR	C	2	Crushing	8.2	13.5
						Crushing	2.5	4.2
						Debonding and crushing	5.3	7.2
Faella et al., 2010 [38]	DG	9	TF	C	1	Debonding and crushing	5.7	---
Augenti et al., 2011 [39]	CY	1	BR	G	1	Tensile rupture	1.03	2.8
Parisi et al., 2013 [40]	DG	6	TF	G	1/2	Tensile rupture	2.4	2.7
Ismail and Ingham, 2016 [29]	CY	1	BR	S	1	Tensile rupture	1.2	1.2
Marcari et al., 2017 [32]	DG	3	TF	B	1	Vertical cracks	1.3	2.4
Marcari et al., 2017 [32]	DG	2	TF	B	2	Vertical cracks	1.6	1.6
Sagar et al., 2017 [31]	DG	15	BR	G	1	Tensile rupture	1.03	7.2
Longo et al., 2021 [41]	DG	2	BR	G	2	Vertical cracks	1.6	-
Longo et al., 2021 [41]	DG	2	BR	S	2	Tensile rupture	1.7	-
Donnini et al., 2021 [42]	DG	3	TF	G	2	Vertical cracks	2.7	4.0
Donnini et al., 2021 [42]	DG	3	BR	G	2	Vertical cracks	1.3	1

**Table 2 materials-14-07021-t002:** Physical and mechanical characteristics of the strengthening mortar (provided by the manufacturer) [47].

Consistency of mix	Plastic–thixotropic
Density of wet mix (kg/m^3^)	1850
Coating	Epoxy–Silicone–PFT–Titanium
Workability (h)	1
Water absorption (kg/(m^2^·min^0.5^)	≤0.1
Thermal conductivity (W/m·K)	0.73
Compressive strength (MPa)	25 (after 28 days)
Flexural strength (MPa)	8 (after 28 days)
Compressive modulus of elasticity (MPa)	10,000 (after 28 days)
Initial shear strength (MPa)	0.15

**Table 3 materials-14-07021-t003:** Physical and mechanical characteristics of the mesh (provided by the manufacturer) [48].

Type of Fiber	High-Strength Carbon Fiber
Weight (g/m^2^)	170
Mesh size (mm)	10 × 10
Density (kg/m^3^)	1830
Tensile strength (MPa)	4800
Modulus of elasticity (MPa)	230,000
Load-resistant area per unit of width (mm^2^/m)	48
Equivalent thickness of dry fabric (mm)	0.048
Elongation at failure (%)	2

**Table 4 materials-14-07021-t004:** Physical and mechanical characteristics of the cords (provided by the manufacturer) [51].

Type of Fiber	High-Strength Carbon Fiber
Appearance	cord
Density (kg/m^3^)	1800
Equivalent surface area of dry fabric (mm^2^)	21.24
Modulus of elasticity (MPa)	230,000
Tensile strength (MPa)	4830
Elongation at failure (%)	2

**Table 5 materials-14-07021-t005:** Physical and mechanical characteristics of the ɸ8 mm cords (provided by the manufacturer) [52].

Appearance	Thixotropic Paste
Density (kg/m^3^)	1690
Compressive strength (MPa)	68
Flexural strength (MPa)	30
Flexural dynamic modulus of elasticity (MPa)	4025
Compressive modulus of elasticity (MPa)	6105

**Table 6 materials-14-07021-t006:** Summary of experimental and numerical results.

Characteristics	URM	TSM	TRM1	TRM2	TRM3
P_ult_exp_ (kN)	25.432	54.378	43.024	58.695	59.364
P_ult_num_ (kN)	24.900	57.210	45.155	58.345	58.345
S_s_exp_ (MPa)	0.043	0.157	0.114	0.167	0.137
S_s_num_ (MPa)	0.042	0.138	0.114	0.152	0.152
Ɣ_exp_ (mm/mm)	0.007	0.017	0.012	0.015	0.012
Ɣ_num_ (mm/mm)	0.008	0.015	0.015	0.014	0.014
G_exp_ (MPa)	6.142	9.235	9.500	11.133	11.417
G_num_ (MPa)	5.250	9.200	7.600	10.857	10.857
E_exp_ (MPa)	15.355	23.088	23,750	27.833	28.542
E_num_ (MPa)	13.125	23.000	19.000	27.143	28.843
δ_u_exp_ (%)	0.220	0.609	0.473	0.650	0.331
δ_u_num_ (%)	0.236	0.884	0.707	0.707	0.707

## Data Availability

The data underlying this article will be shared on reasonable request from the corresponding author.

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
