# Peer review of "Diagonal Tensile Test on Masonry Panels Strengthened with Textile-Reinforced Mortar"

_materials, 2021, doi:10.3390/ma14227021_

Round 1
Reviewer 1 Report
- The number of specimens is inconsistent. Reference needs minimum two specimens in order to compute and average result. Authors are invited to deeply comment this aspect.
- Table 1 is very limited database if compare with the available literature. Recent investigation evidenced also the positive attitude of TRM for both seismic and energy improvement. Please consider the following “Longo, F., Cascardi, A., Lassandro, P., & Aiello, M. A. (2021). Energy and seismic drawbacks of masonry: a unified retrofitting solution. Journal of Building Pathology and Rehabilitation, 6(1), 1-24.”
- Please provide coefficient of variation when computing the average result.
- Please avoid commercial names.
- The influence of the type o transversal connection r was not commented. Authors are invited to consider the following “Cascardi, A., Leone, M., & Aiello, M. A. (2020). Transversal joining of multi-leaf masonry through different types of connector: Experimental and theoretical investigation. Construction and Building Materials, 265, 120733.”
- Please improve fig 16. The structure seems to be labile.
- Please use strength vs strain instead of load vs displacement.
- Conclusions are too long
Author Response
The authors would like to thank the reviewer for the careful and thorough reading of this manuscript and for the thoughtful comments and constructive suggestions that help to improve the quality of this paper. We have incorporated all the suggestions made by the reviewer. Those changes are highlighted within the manuscript. Please see below a point-by-point response to the reviewers’ comments and concerns. All page numbers refer to the revised manuscript file with tracked changes.
The number of specimens is inconsistent. Reference needs minimum two specimens in order to compute and average result. Authors are invited to deeply comment this aspect.
- Three masonry panels (TRM1, TRM2 and TRM3) were strengthened by TRM plastering. The unstrengthen and the traditionally strengthen masonry panels were used as benchmark (URM and TSM). Thus, the experimental and numerical results that were obtained for these two panels were used as a point of reference for the evaluation of the structural performances of the three TRM strengthened panels.
Table 1 is very limited database if compare with the available literature. Recent investigation evidenced also the positive attitude of TRM for both seismic and energy improvement. Please consider the following “Longo, F., Cascardi, A., Lassandro, P., & Aiello, M. A. (2021). Energy and seismic drawbacks of masonry: a unified retrofitting solution. Journal of Building Pathology and Rehabilitation, 6(1), 1-24.”
- We agree with the reviewer’s assessment. Accordingly, throughout the manuscript, we have revised and indicated in Table 1 more recent articles related to the experimental campaigns performed on strengthen masonry panels. Moreover, the authors consider that the article indicated by the reviewer provided valuable information and it is a good point of reference to compare the results of our work. Thus, this article was also cited.
Please provide coefficient of variation when computing the average result.
- The experimentally determined values for the compressive strength of bricks and tensile and compressive strengths of the masonry mortar and strengthening mortar are now presented into text. For each set of data, the standard variation was computed and indicated in the text (see lines 147-155; 164-175; 205-212)
Please avoid commercial names.
- Thank you for pointing this out. We have revised and eliminated the commercial names.
The influence of the type o transversal connection r was not commented. Authors are invited to consider the following “Cascardi, A., Leone, M., & Aiello, M. A. (2020). Transversal joining of multi-leaf masonry through different types of connector: Experimental and theoretical investigation. Construction and Building Materials, 265, 120733.”
- In the revised version, the influence of the transversal connectors was commented (see lines 463-473). Regarding the shape of the connector, the explanations were given with respect to the indicated article (see citation 66).
Please improve fig 16. The structure seems to be labile.
- The figure was improved.
Please use strength vs strain instead of load vs displacement.
- The load vs displacement distribution curves are presented in figures 16, 18, 21, 22, 23 while the shear stress vs shear strain distribution are presented in figures 24 and 25.
Conclusions are too long
- Thank you for pointing this out. We have revised and condensed the conclusion section.
Reviewer 2 Report
The paper, entitled “Shear Strengthening of Unreinforced Masonry panels with textile reinforced mortar” reports on the results of an experimental and numerical program carried out on unreinforced masonry panels strengthened by textile reinforced mortar (TRM) plastering.
The results are interesting; however, the structure of the paper needs improvements. The materials and methods section should be revised by deleting redundant information, and by clarifying the type of series considered and materials adopted. The results are presented in a chaotic manner; moreover, the numerical analysis should be presented separately from the experimental one. Finally, the discussion of the influence of the reinforcement on main parameters, such as capacity, stiffness, ductility, is not properly addressed. In the Reviewer’s opinion major revisions are needed to make the paper suitable for publication, according to the comments listed as follow:
- The State of the Art is widely and deeply described in the Introduction section. However, no mention is done with respect to the TRM characterisation methods by means of tensile test and shear bond test with the substrate. Before mentioning the lack of studies with respect to anchorage (line 102), it would be appropriate to refer to this aspect. You can find as follow an example of studies moving from the local characterisation of TRM (tensile and adherence tests) to the global characterisation by means of diagonal compression tests on reinforced masonry walls (it’s just an example, you can find plenty of similar works in literature)
-G. Ferrara, E. Martinelli, Tensile behaviour of Textile Reinforced Mortar composite systems with flax fibres, in: Proc. of the 12th fib International PhD Symposium in Civil Engineering, 29, 2018, pp. 1–7.
- G. Ferrara, C. Caggegi, E. Martinelli, A. Gabor, Shear capacity of masonry walls externally strengthened using Flax-TRM composite systems: experimental tests and comparative assessment, Construct. Build. Mater. 261 (2020) 120490, https://doi. org/10.1016/j.conbuildmat.2020.120490.
- Table 3 is cited before Table 2 in the text. Please make them in the correct order.
- Lines 138-149. It is clear the characterisation of the bricks occurred according to the cited Standard, However, the more details should be provided on the method: testing machine, number of bricks tested, number of tests per brick. Figure 4 is probably redundant (It would be enough to insert standard deviations in Table 3). If more details are not provided, then is better to just mention the standard and the deriving compression strength.
- Lines147-150. “It is thus ensured that the numerical analysis simulates the most unfavorable case in terms of mechanical properties of the URM panel”. Which is the aim of your numerical model? If the aim is to reproduce the experimental tests, then the actual assessed strength should be adopted. If the aim is to provide a numerical model for designing purposes, it could make sense to use conservative values. Please, explain the purpose of the proposed numerical model.
- Lines 158-184. In my opinion it is redundant to show so in detail the mortar characterisation. It could be enough just citing the standard of reference and the deriving strength values (I would do this for both bricks and mortar characterisation). What is missing is the origin of the mortar samples: from which wall the mortar samples were extracted? Strictly speaking, for each mortar batch adopted for the wall casting the mechanical characterisation should be provided.
- Lines 186-194. It is not clear which series of specimen were considered. Please clarify in detail the characteristic of each wall, also by assigning them a name. For instance URW unreinforced wall, TRM_RW wall strengthened by TRM system, TRM_RW_A wall strengthened by TRM system with anchors, etc…
- Lines 197-201. Again, too confusing. List together all the series with their parameters (either by making a list or using a dedicated table).
- In my opinion the fibres and TRM adopted should be described before the description of the reinforced walls, so it would be easier to understand the reinforcement system adopted for the specific wall.
- “…strengthened by plastering on one face (for one specimen) or on both faces (for two specimens)..” Again, too confusing. Please elaborate again this section according to previous comments.
- Lines 207-218: It could be enough just citing the standard of reference and the deriving strength values. Fig 10-11-12 and table 7 and 8 are redundant.
- “For the micro non-linear 3D model, the mechanical properties were defined according to the experimentally deter-227 mined values” this choice is not consistent with lines 147-150. Please justify it.
- Lines 243-254. These information should be placed in the State of the art section, not in the Materials and Method section.
- Lines 260-266. Please spcify the amount and the position of the connectors (for instance also by detailing Figure9).
- Line 283. Please specify the proportion among water, sand and binder.
- Lines 289-294. Please specify the overall thickness of the reinforcement layer.
- Section 3.2: please specify the mechanical parameters adopted for the considered elements (i.e. stiffness, strength).
- Line 447. Please, before referring to Figure 21, specify that the results are expressed in terms of Load, vertical/horizontal displacements (in front and back of the walls) and also in terms of numerical response.
- Lines 456-466 is the repetition of lines 443-453.
- “A similar behavior was recorded for the second reference panel, the traditional strengthened specimen (Figures 22, 23).” What do you mean for similar behaviour? From the picture it is clear the failure mode significantly involves the reinforcing matrix, while the Unreinforced reference wall does not even have a reinforcing matrix. Please explain better what do you mean.
- Load displacement curves: Please use the same scales (for Y and X axis) for all the results so it is easy to compare them.
- As general comment the result and discussion section may be clearer if organised in different way: As first showing the results (load displacement curves), then discussing the main parameters (stiffness, maximum capacity, ductility, etc…), then showing the failure mode. Moreover, the results in terms of numerical model probably deserve a dedicated section in which they are discussed in detail.
- Fig 34 and 35 show the same curves of figures 29-33. Showing them 2 times is redundant. In my opinion the best effective way to show the results is the one proposed in Fig 34 and 35.
- Lines 566-576. The parameters G, E, delta u, need to be properly discussed for all the series of specimens. It is important understanding the influence of the reinforcing system in terms of ductility, stiffness and capacity.
Author Response
The authors would like to thank the reviewer for the careful and thorough reading of this manuscript and for the thoughtful comments and constructive suggestions that help to improve the quality of this paper. We have incorporated all the suggestions made by the reviewer. Those changes are highlighted within the manuscript. Please see below a point-by-point response to the reviewers’ comments and concerns. All page numbers refer to the revised manuscript file with tracked changes.
The State of the Art is widely and deeply described in the Introduction section. However, no mention is done with respect to the TRM characterisation methods by means of tensile test and shear bond test with the substrate. Before mentioning the lack of studies with respect to anchorage (line 102), it would be appropriate to refer to this aspect. You can find as follow an example of studies moving from the local characterisation of TRM (tensile and adherence tests) to the global characterisation by means of diagonal compression tests on reinforced masonry walls (it’s just an example, you can find plenty of similar works in literature)
- We agree with the reviewer’s assessment. Accordingly, throughout the manuscript, we have revised and indicated in more recent studies related to the TRM characterization methods by means of tensile test and shear bond test with the substrate. Please see lines 94-98; 100-106 and citations 35, 36, 43.
Table 3 is cited before Table 2 in the text. Please make them in the correct order.
- The text was revised and the citations were rearranged.
Lines 138-149. It is clear the characterisation of the bricks occurred according to the cited Standard, However, the more details should be provided on the method: testing machine, number of bricks tested, number of tests per brick. Figure 4 is probably redundant (It would be enough to insert standard deviations in Table 3). If more details are not provided, then is better to just mention the standard and the deriving compression strength.
- Figure 4 was removed. The standard deviation was introduced in the text. More information related to the compressive tests (number of specimens, testing machine, standard deviation, mean value, etc) were presented (see lines 147-155).
Lines147-150. “It is thus ensured that the numerical analysis simulates the most unfavorable case in terms of mechanical properties of the URM panel”. Which is the aim of your numerical model? If the aim is to reproduce the experimental tests, then the actual assessed strength should be adopted. If the aim is to provide a numerical model for designing purposes, it could make sense to use conservative values. Please, explain the purpose of the proposed numerical model.
- The aim of the numerical analysis was indicated (see lines 156-159).
Lines 158-184. In my opinion it is redundant to show so in detail the mortar characterisation. It could be enough just citing the standard of reference and the deriving strength values (I would do this for both bricks and mortar characterisation). What is missing is the origin of the mortar samples: from which wall the mortar samples were extracted? Strictly speaking, for each mortar batch adopted for the wall casting the mechanical characterisation should be provided.
The mortar samples were not extracted. Instead, prism samples from mortar type S, with the nominal dimensions of 40 x 40 x 160 mm were prepared and tested (see figure 4). Since the masonry panel were manufactured and tested in the laboratory, there was no relevance to extract and test a different type of mortar from an existing wall.
- The mortar samples were not extracted. Instead, prism samples from mortar type S, with the nominal dimensions of 40 x 40 x 160 mm were prepared and tested (see figure 4). Since the masonry panel were manufactured and tested in the laboratory, there was no relevance to extract and test a different type of mortar from an existing wall.
- The redundant information (figures and tables) were removed. The mean value and the standard deviation were introduced in the text (see lines 164-175).
Lines 186-194. It is not clear which series of specimen were considered. Please clarify in detail the characteristic of each wall, also by assigning them a name. For instance URW unreinforced wall, TRM_RW wall strengthened by TRM system, TRM_RW_A wall strengthened by TRM system with anchors, etc…
- To facilitate the interpretation of the results, codes were introduced for each masonry panel (URM -unreinforced panel, TSM – traditional strengthened panel, TRM1 – panel strengthened on one face, TRM2 and TRM3 – panels strengthened on both faces). Please see lines 181-183; 192-195.
Lines 197-201. Again, too confusing. List together all the series with their parameters (either by making a list or using a dedicated table).
- To facilitate the interpretation of the results, codes were introduced for each masonry panel (URM -unreinforced panel, TSM – traditional strengthened panel, TRM1 – panel strengthened on one face, TRM2 and TRM3 – panels strengthened on both faces).
In my opinion the fibres and TRM adopted should be described before the description of the reinforced walls, so it would be easier to understand the reinforcement system adopted for the specific wall.
- In the experimental se-up section, there were described the unreinforced panel, the traditional reinforced panel and the textile reinforced ones. For each type of panel, there were indicated the configuration, the materials and the manufacturing technique. The properties of the fibres and mortar were described immediately after the geometrical configurations of the TRM panels (225-239).
“…strengthened by plastering on one face (for one specimen) or on both faces (for two specimens)..” Again, too confusing. Please elaborate again this section according to previous comments
- The text was revised, according to the previous indication (see lines 192-195).
Lines 207-218: It could be enough just citing the standard of reference and the deriving strength values. Fig 10-11-12 and table 7 and 8 are redundant.
- The redundant figures and tables were deleted
- The mean value and the standard deviation were introduced in the revised text (see lines 147-155; 164-171; 205-212).
“For the micro non-linear 3D model, the mechanical properties were defined according to the experimentally deter-227 mined values” this choice is not consistent with lines 147-150. Please justify it
- Yes, it is. If a numerical model is aimed to be utilize for design purposes, the input data in terms of strength and stiffness should be based on the conservative values.
- For the bricks, the conservative values (in terms of mechanical characteristics) were the ones provided by the manufacturer, while for the strengthening mortar, the conservative values were the experimentally determined ones.
- Please see the revised text from the lines 205-2015.
Lines 243-254. These information should be placed in the State of the art section, not in the Materials and Method section
- This information is related directly to the specific materials that were used in this work. Neither there are results obtained by previously experimental work nor general characteristics valid for any type of meshes and mortars.
Lines 260-266. Please spcify the amount and the position of the connectors (for instance also by detailing Figure9)
- The text was revised. Please see lines 242-245.
Line 283. Please specify the proportion among water, sand and binder.
- The text was revised (see lines 263-265)
Lines 289-294. Please specify the overall thickness of the reinforcement layer.
- The text was revised (see lines 274-275)
Section 3.2: please specify the mechanical parameters adopted for the considered elements (i.e. stiffness, strength).
- In section 2 – Experimental se-up there were indicated all the properties of the constituent materials (both the ones indicated by the manufacturer and the ones that were experimentally determined). For each case there was indicated which data were adopted for the numerical model.
Line 447. Please, before referring to Figure 21, specify that the results are expressed in terms of Load, vertical/horizontal displacements (in front and back of the walls) and also in terms of numerical response.
- The text was revised. Please see lines 422-431.
Lines 456-466 is the repetition of lines 443-453.
- Thank you for pointing this out. The duplication was removed.
“A similar behavior was recorded for the second reference panel, the traditional strengthened specimen (Figures 22, 23).” What do you mean for similar behaviour? From the picture it is clear the failure mode significantly involves the reinforcing matrix, while the Unreinforced reference wall does not even have a reinforcing matrix. Please explain better what do you mean.
- The text was revised and the misunderstanding was removed (see lines 442-449).
Load displacement curves: Please use the same scales (for Y and X axis) for all the results so it is easy to compare them.
- The limits of the scales were set in such a way that the general response of the panels is presented in the most illustrative way and also by also taking into account the size limitations which are imposed by the journal’s format. We are kindly recommending to present the figures with the scales provided by the authors in order to better understand the behavior of the masonry panels.
As general comment the result and discussion section may be clearer if organised in different way: As first showing the results (load displacement curves), then discussing the main parameters (stiffness, maximum capacity, ductility, etc…), then showing the failure mode. Moreover, the results in terms of numerical model probably deserve a dedicated section in which they are discussed in detail.
- In some articles, the results are organized in the manner indicated by the reviewer. However, in other articles, the results are addressed in the same order as in this work. It is common to present a picture with the failed specimen and below this picture to characterize the failure mode, to indicate the ultimate values of force, displacement, stress, strain, etc, and their variations. It is also common to present comparative graphs in terms of numerical and experimental data. In this manner it is easy to identify the limitation of a numerical model and to have a general view of the structural performances of the specimens.
Fig 34 and 35 show the same curves of figures 29-33. Showing them 2 times is redundant. In my opinion the best effective way to show the results is the one proposed in Fig 34 and 35.
- The curves were displayed in the manner indicated by the reviewer (see figures 24 and 25).
Lines 566-576. The parameters G, E, delta u, need to be properly discussed for all the series of specimens. It is important understanding the influence of the reinforcing system in terms of ductility, stiffness and capacity.
- Thank you for pointing this out. We have revised and introduced the information (please see lines 548-566).
Reviewer 3 Report
The paper presents an experimental and numerical study on the strengthening of URM with TRM to improve the diagonal cracking strength. The paper fits within the scope of the journal and is generally well written. The reviewer suggests revision of the paper according to the following comments:
- The title of the paper is inaccurate. Diagonal tension tests are not typical shear loading tests for URM. Please modify the title accordingly.
- Tables 1 and 2 could be integrated. There is no reason to present all different values of brick specimens. The mean value and the standard deviation are adequate.
- Figure 3 has to be replaced by a close up photo.
- Figure 4 could be deleted. It is not important.
- Tables 3 and 4 can be integrated according to the comment No. 1.
- Figures 6 and 7 can be deleted. They do not provide useful information. Standard deviation can be described in the text.
- Same comment for Figures 11, 12 and Tables 7 and 8.
- Are the fibers of the textile mesh coated or dry? Please add this information in the text.
- How did the authors select the configuration of the anchors? Please add the relevant information.
- It is strange to see a "plastic" behaviour of the URM panels in such kind of tests. One would expect sudden loss of the capacity and spitting of the wall in two parts. Lines 456-465 need to be revised to better describe the behaviour of the walls along with pictures.
- Table 12: How is it possible to get lower stiffness in strengthened URM compared to the unstrengthened URM? This cannot be right.
Author Response
The authors would like to thank the reviewer for the careful and thorough reading of this manuscript and for the thoughtful comments and constructive suggestions that help to improve the quality of this paper. We have incorporated all the suggestions made by the reviewer. Those changes are highlighted within the manuscript. Please see below a point-by-point response to the reviewers’ comments and concerns. All page numbers refer to the revised manuscript file with tracked changes.
The title of the paper is inaccurate. Diagonal tension tests are not typical shear loading tests for URM. Please modify the title accordingly.
- The title was revised.
Tables 1 and 2 could be integrated. There is no reason to present all different values of brick specimens. The mean value and the standard deviation are adequate.
- The mean value and the standard deviation were introduced in the revised text (see lines 164-171). The tables were deleted.
Figure 3 has to be replaced by a close up photo.
- The figure was replaced.
Figure 4 could be deleted. It is not important.
- Figure 4 was deleted.
Tables 3 and 4 can be integrated according to the comment No. 1.
- The tables were deleted and the data were introduced in the text (see lines 170-175)
Figures 6 and 7 can be deleted. They do not provide useful information. Standard deviation can be described in the text
- The figures were deleted. Standard deviation was described in the text
Same comment for Figures 11, 12 and Tables 7 and 8.
- The figures and the tables were deleted. The data were introduced in the text (see lines 205-215).
Are the fibers of the textile mesh coated or dry? Please add this information in the text.
- The information was added (see lines 226-227).
How did the authors select the configuration of the anchors? Please add the relevant information.
- The information was added (see lines 248-249).
It is strange to see a "plastic" behaviour of the URM panels in such kind of tests. One would expect sudden loss of the capacity and spitting of the wall in two parts. Lines 456-465 need to be revised to better describe the behaviour of the walls along with pictures.
- The text was revised (see lines 418-431).
Table 12: How is it possible to get lower stiffness in strengthened URM compared to the unstrengthened URM? This cannot be right.
- Thank you for pointing this out. It was an error. We have revised. Please see Table 4 and lines 548-566.
Round 2
Reviewer 1 Report
The paper was significantly improved and can be now accepted in the present form
Reviewer 2 Report
The authors addressed all comments adequately. Publication is suggested.
Reviewer 3 Report
The authors addressed all comments adequately. Publication is suggested.